

# Laboratory observations on meltwater meandering rivulets on ice

Roberto Fernández[1, 2], and Gary Parker[2,3]

[1]Energy and Environment Institute, University of Hull, Hull, HU6 7RX, UK
[2]Ven Te Chow Hydrosystems Laboratory, Department of Civil and Environmental Engineering, University of Illinois at Urbana-Champaign, Urbana, IL 61801, USA
[3]Department of Geology, University of Illinois at Urbana-Champaign, Urbana, IL 61801, USA

*Correspondence to*: Roberto Fernández (R.Fernandez@hull.ac.uk)

## Abstract

We present a set of observations on meltwater meandering rivulets on ice and compare them (qualitatively and quantitatively) to morphologies commonly found in meandering channels in different media. The observations include data from planned centimeter-scale experiments, and from incidental self-formed millimeter-scale rivulets. Our data show pulsed lateral migration features, undercut banks and overhangs, meander bend skewness, and meander bend cutoffs. The data also compare well with planform characteristics of alluvial meandering rivers (sinuosity, wavelength-to-width ratios, and meander bend fatness and skewness). We discuss the (ir)relevance of scale in our experiments, which in spite of being in the laminar flow regime, and are likely affected by surface tension effects, are capable of shedding light into the processes driving formation and evolution of supraglacial meltwater meandering channels. Our observations suggest that sinuosity growth in meltwater meandering channels on ice is a function of flow velocity and the interplay between vertical and lateral incision driven by temperature differences between flow and ice. In the absence of recrystallization (depositional analog to alluvial rivers), bends are more likely to be downstream skewed and channels show lower sinuosities.

## 1 Introduction

Rivers and other channels containing liquid flow develop meandering patterns over different media and across a broad range of scales. The quintessential example of a meandering channel is the alluvial river, which forms and evolves by erosion and deposition of sediment (e.g. Fisk, 1944; Seminara, 2006) and interaction with riparian vegetation (e.g. Braudrick et al., 2009). Meandering channels also form in bedrock where the driving processes include dissolution (e.g. Veress and Tóth, 2004; Allen, 1971; Zeller, 1967), abrasion by sediment particles (Sklar and Dietrich, 2004), plucking of bedrock (Chatanantavet and Parker, 2009; Whipple et al., 2000), slaking (Johnson and Finnegan, 2015), and weathering (Pelletier and Baker, 2011).

Rivulets on a plane can also meander due to an instability associated with surface tension (Le Grand Piteira et al., 2006; Davies and Tinker, 1986; Culkin and Davis, 1984; Goricky, 1973), and channelized flows over ice surfaces also develop meandering patterns due to differential melting (e.g. Karlstrom et al. 2013; Parker, 1975; Ferguson, 1973; Zeller, 1967). Compared with





the cases of alluvial and bedrock meandering channels, the study of meltwater meandering channels on ice has received little attention (Pitcher and Smith, 2019), and no experimental work has been reported to date.

We present a set of experimental observations which were motivated by the image in Figure 1. It shows an ice island with melt ponds and meltwater meandering channels on its surface. It started as a very large iceberg which broke off Peterman Glacier, and had been splitting into pieces as it drifted south over the Atlantic Ocean for more than a year. This led to our initial question: (i) Can we produce meltwater meandering channels over ice in the laboratory? Simply put, the image shows an ice block with water flowing over it in a purely melting environment. Such conditions were easily reproduced in the laboratory. After successfully dealing with that question, we proceeded to work on answering the following: (ii) Can we reproduce the observations made by previous researchers through field, theoretical, and high-resolution imagery-based analyses, albeit at a much smaller scale? (iii) How do small-scale laboratory meltwater meandering channels on ice compare to their supraglacial relatives and with meandering channels in other media?

## 1.1 Background

Interest in supraglacial meandering channels seems to have been triggered by Leopold and Wolman (1960), who described two meltwater meandering channels in the Dinwoody Glacier, Wyoming and compared their planform and channel geometry to that of alluvial rivers. Similarities between alluvial meanders and supraglacial meltwater meandering channels was confirmed by other authors (Dahlin, 1974; Dozier, 1974; Ferguson 1973; Knighton, 1972; Zeller, 1967). Notwithstanding these similarities, an important difference with alluvial meandering streams was observed.

Supercritical flow conditions were documented by Dahlin (1974) and Knighton (1972), and a linear stability analysis of the problem suggested that they were necessary for meandering to occur in supraglacial streams (Parker, 1975). Nevertheless, sub- and supercritical flow conditions were observed in the Juneau Ice Field (Marston, 1983) and Karlstrom et al. (2013) presented a new theoretical model of meander formation in supraglacial streams in which, among other outcomes, they show that meltwater meandering can occur under both super- and subcritical flow conditions, as long as the Froude number is greater than ~0.4.

Recent research in supraglacial channels has not focused on the issue of meandering but rather on aspects such as channel inception (e.g. Mantelli and Camporeale, 2015), river networks (e.g. Yang et al. 2016), controls on drainage development and pathways (e.g. Banwell et al., 2016; Rippin et al., 2015), and supraglacial landscape evolution (e.g. Karlstrom and Yang, 2016). In their review of supraglacial streams and rivers, Pitcher and Smith (2019) discuss the issue of meandering briefly and note that in the past, the problem has been addressed with theoretical approaches (Karlstrom, 2013; Parker, 1975), field observations (Marston 1983; Hambrey 1977; Dozier 1976, 1974; Ferguson 1973; Knighton 1972; Zeller 1967; Leopold and Wolman 1960) and, with the advent of high-resolution imagery, remote sensing (Rippin et al., 2015).

In this paper we (i) describe the experimental approach leading to the successful development of cm-scale and mm-scale meltwater meanders in the laboratory; (ii) present a qualitative, and a simple quantitative comparison (including sinuosity, wavelength-to-width ratio, skewness, and fatness), of the small scale meltwater meandering rivulets and meandering channels



in other media; and (iii) interpret the differences and similarities observed in light of the different mechanisms driving the meandering processes.

## 2 Materials and Methods

This section provides a brief description of the methods used to conduct the experiments, process and analyze the data. A more comprehensive description is provided as supplemental material (S1), and is also available in Fernández (2018).

### 2.1 Centimeter Scale Experiments

Experiments were conducted over 0.2 m thick ice blocks with surface area of 0.96 m by 0.38 m. A rivulet was carved over the ice block with a chisel. Elevations along the carved rivulet and the ice surface were measured with a point gage. This was done before the run and at different times during the run to measure channel slope. A constant flow rate of 0.25 liters per minute (4.2 cm³s⁻¹) was used for all runs. Figure 2 shows the basic setup for the cm-scale experiments.

Planform evolution of the channel was documented with time lapse imagery. Images were acquired at a constant frequency with typical time steps of 2-3 seconds depending on the experiment, although we report on a run that had a time step of 10 seconds. Inflow water temperature was measured with a common thermometer, and ice block temperatures were measured with a handheld infrared thermometer. Approximate flow depths were measured with a periodontal probe during the run and also estimated afterwards as follows: $H = Q/(UB)$ where H is depth, Q is flow discharge, U is reach-averaged velocity, and B is reach-averaged channel width.

The experimental measurements were complemented with a mold of the channel made with a Room-Temperature-Vulcanizing (RTV) Silicone. The silicone rubber was poured into the channel and allowed to harden while the ice block melted. The mold was recovered once the ice block melted completely.

### 2.2 Millimeter Scale Experiments

The mm-scale experiments were, at first, an unexpected incidental by-product of the cm-scale experiments. As the channel resulting from a cm-scale run was being molded, the mm-scale channels formed from thin meltwater flow generated on the surface of the ice block to either side of the mold, by heat release from molding material as it hardened. Figure 3 shows the first set of mm-scale channels that was observed. Following this discovery, the formative process was intentionally repeated to document the planform characteristics of the mm-scale channels.

Subsequent runs involved molding a cm-scale channel and letting the mm-scale channels form and evolve. The small size of the mm-scale channels only allows measuring planform morphology with the help of images. This was achieved by adding dye over the ice surface once the channels were formed.





### 2.3 Image Analysis and Planform Characteristics

Channel centerlines were manually digitized in Matlab and all length scales measured in the images were converted to real length units using the corresponding resolution. The analysis includes seven centerlines from the Peterman Ice Island, seventeen mm-scale channels, and four cm-scale runs from which twenty six centerlines were extracted as the channel evolved
in time. Supplemental material includes images showing the mm-scale and ice-island centerlines (S2) and videos showing two cm-scale runs are also available (Video supplement). The parameters used to describe the planform characteristics of the meltwater meanders are channel sinuosity, wavelength-to-width ratio, coefficient of fatness, and coefficient of skewness (Parker et al., 1982).

Langbein and Leopold (1966) noted that meander bends sometimes possess double-valued planforms which makes them look
round and full, or as Parker et al. (1982) put it, 'fat'. The coefficient of fatness is a measure of how round or angular a meander bend is. Skewness refers to the direction towards which a meander bend is pointing with respect to the direction of the flow. Figure 4 shows a graphic description of these parameters. The top portion of the figure shows a round meander with a positive fatness coefficient ($f_f > 0$) and an angular meander with a negative fatness coefficient ($f_f < 0$). The middle portion of the figure shows a downstream-skewed bend with a positive skewness coefficient ($f_s > 0$) and an upstream-skewed bend with a
negative skewness coefficient ($f_s < 0$). Upstream skewness coefficients are negative because all channel centerline coordinates in this study were defined such that they are positive in the downstream direction. Figure 4 also shows a definition sketch for meander sinuosity. Sinuosity ($\Omega$) is the ratio between the along-channel distance and the valley (straight) distance as follows:

$$\Omega = (Dist_{ABCDE}) \cdot (Dist_{AE})^{-1} \tag{1}$$

Each channel centerline digitized in Matlab was smoothed (Güneralp and Rhoads, 2008), and its direction spatial series was
standardized so that its average value over the entire reach was zero (Zolezzi and Güneralp, 2015). Meander bend wavelengths, and coefficients of fatness and skewness, were computed with 'meanderscribe', a set of Matlab scripts developed by Vermulen et al. (2016) to study the multiscale structure of meander bends. This procedure was also applied to rivers in the NCHRP database (Lagasse et al., 2004) so as to compare our results with the rivers therein. Specific details about the extraction and processing of the NCHRP data are included in the supplemental material (S1).

The time-lapse images from the cm-scale runs were used to measure median channel widths, reach-averaged flow velocities, and lateral migration rates. Median channel widths were determined by measuring widths along the centerline at various locations and computing the median value. Reach-averaged velocities were determined by calculating the along-centerline distance traveled by injected dye fronts in subsequent images and dividing by the time between them. Lateral migration rates were determined by computing the displacement of the apex of a given meander bend at different times throughout the run
and dividing by the corresponding time interval.



## 3 Results

### 3.1 Signatures of lateral migration and vertical incision

Pulsed lateral migration features (Figure 5), undercut banks and overhangs (Figure 6), meander bend skewness (Figure 7), and meander bend cutoffs (Figure 8), are common signatures of lateral migration in meandering channels. The timescales between

formation and obliteration of these signatures are a function of channel medium and scale. All four signatures are present in our experimental observations. The figures in this section show a qualitative comparison with such signatures in other environments.

Pulsed lateral migration episodes, analogous to those responsible for the formation of scroll bars and terraces, were recorded as streak lines on the overhangs on the outside of bends in the cm-scale experiments (Fig. 5c). Persistence of these features

was a function of ice block melting rates and the depth within the ice block at which they formed. Smooth sloping terraces similar to alluvial point bars formed on the inner side of bends during the experiments (Fig. 6c, 7c).

In the case of alluvial rivers, scroll bars reflecting lateral migration pulses are found on the inner bank (Fig. 5a) (e.g. Strick et al., 2018). On the outer bank it is possible to observe undercut banks and overhangs (Fig. 6a) (e.g. Thorne and Tovey, 1981). Erosion of the bank toe leads to cantilever failure of the overhang and to the formation of slump blocks (e.g. Hackney et al.,

2015). We did not observe overhang collapse due to thermal erosion at the toe of the outer bank in our experiments, but the uppermost layers of the overhangs melted away as the channel incised deeper into the ice block and the ice block surface melted.

In bedrock meandering rivers, lateral migration signatures are observed as terraces on the inner parts of bends (Fig. 5b, 7a) (e.g. Johnson and Finnegan, 2015; Finnegan and Dietrich, 2011), and undercuts and overhangs on the outer parts of bends

(Fig. 6b) (e.g. Inoue et al., 2017). Terraces are also found in supraglacial meltwater meandering channels (Fig. 7b). In smaller scale channels, these terraces on the inner parts of bends melt according to the overall glacial surface ablation rates. In contrast with the bedrock case, their elevation with respect to the bed of the channel remains relatively constant (Karlstrom et al. 2013; Knighton, 1981). We presume this was also occurring in the mm-scale experiments since the elevation difference between channel thalweg and ice block surface did not seem to change in time. In the cm-scale experiments the terraces more closely

resembled the case of bedrock meandering terraces (e.g. Fig. 7a, 7d). Vertical incision rates were much faster than ice block melting rates, thus leaving terraces high above the channel thalweg.

Skewed bends also appeared in both sets of experiments (Fig. 7c). Skewness direction (upstream or downstream) depends on local processes and conditions driving planform evolution. Figure 7 shows four examples of meandering channels with very different scales; all show skewed bends. Lateral migration promotes meander bend growth, thus increasing sinuosity. The

process is, nevertheless, regulated by the occurrence of cutoffs (Fig. 8). Figure 8 shows examples of chute and neck cutoffs in alluvial, bedrock, supraglacial meltwater rivers, and meltwater laboratory rivulets. An imminent neck cutoff in a millimeter scale channel is also shown in Figure 7c.



In alluvial meandering rivers, cutoffs leave abandoned bends whose inlet and outlet become plugged by sediment, preventing flow from entering the bend and forming an oxbow lake (e.g. Constantine and Dunne, 2008). These lakes can fill up with mud during floods and eventually become covered by vegetation (clay plug). In bedrock meandering rivers, abandoned bends might also be observed after cutoffs. Due to the limited sediment supply and continuous vertical incision of the channel itself,

abandoned bends do not fill up and are found at higher elevations with respect to the active channel (Fig. 8c).

In supraglacial meandering channels, cutoffs also lead to the formation of abandoned bends (Rippin et al., 2015). As in the bedrock case, the bends do not become oxbow lakes and remain at a higher elevation than the active channel. Somewhat similar to the alluvial case where sedimentation and vegetation growth might hide the abandoned bend from the naked eye, glacial surface ablation in the warm months and snowfall in the colder months might completely obliterate abandoned supraglacial

meander bends (Rippin et al., 2015). In the mm-scale experiments, abandoned bends were not preserved due to melting of the ice block. In the cm-scale experiments, a neck cutoff was discovered after making the silicone mold of the channel (Fig. 8d). We did not observe the cutoff as it happened because it occurred approximately 0.10 m away from the ice block surface, and the channel had migrated laterally underneath an overhang.

Figure 9 shows examples of knickpoints in the field and in the laboratory. Knickpoints are localized steps in the river profile.

They do not represent a morphological feature specific to meandering streams, but their formation and upstream propagation in our cm-scale experiments is worth noting. Two knickpoints were observed during an experiment and were recorded in a mold (Fig. 9c). The knickpoints migrated upstream as the channel continued to incise vertically into the ice block (Fig. 9e). The neck cutoff shown in Figure 8d is also shown in Figure 9c. We believe that the downstream-most knickpoint and the meander bend cutoff are related. It is likely that the neck cutoff produced a knickpoint, as has been observed in bedrock

meandering rivers (e.g. Finnegan and Dietrich, 2011).

### 3.2 Planform morphology and evolution

Planform morphology of the experimental rivulets was quantified with sinuosity ($\Omega$), wavelength-to-width ratio ($\lambda B^{-1}$), coefficient of skewness ($f_s$), and coefficient of fatness ($f_f$). The results are compared with the seven centerlines extracted from the Peterman Ice Island (Fig. 1), and the alluvial meandering rivers in the NCHRP database (Lagasse et al. 2004). Figure

10 shows boxplots of the results and Table 1 summarizes the statistical values obtained. Figure 11 shows the planform evolution of four examples of cm-scale meandering channels. Table 2 summarizes measured and estimated hydraulic parameters for the four runs at different times during the experiments. It also shows width-to-depth ratios, measured lateral migration rates, and sinuosity values.

In Figure 10, the first and second boxes correspond to the mm- and cm-scale laboratory experiments respectively. The third

box corresponds to the channels extracted from the image of the Peterman Ice Island and the last box corresponds to the rivers in the NCHRP database. Boxes are plotted according to the following rules: the median value is shown as the horizontal line inside the box, and the top and lower edges of the box indicate the 75th and 25th percentiles, respectively. The whiskers extend to the most extreme data points not considered outliers. A value is considered an outlier if it is greater than $q_3 + 1.5(q_3 - q_1)$





or smaller than $q_1 - 1.5(q_3 - q_1)$ where $q_3$ is the 75th percentile (or third quartile) and $q_1$ is the 25th percentile (or first quartile).

### 3.2.1 Sinuosity

Sinuosity values measured in the mm-scale and cm-scale meandering channels are very different from each other (Fig. 10a). The former have a median value of 2.01 whereas the latter have a median value of 1.36 (Table 1). The median sinuosity values measured for the channels over the Peterman Ice Island are 1.50, closer to the cm-scale channels. The median sinuosity obtained for the NCHRP rivers is 1.67, but the data spread in this case actually covers the ranges observed in all other three cases. In the case of alluvial rivers, different values have been reported depending on the geographic area being studied and the number of reaches included in the analysis. For example, MacDonald et al. (1991) report the sinuosity values of 16 reaches of meandering rivers in Minnesota. They observed values between 1.21 and 2.61 with a median value of 1.96, closer to the mm-scale meandering channels.

The low sinuosity values observed in the cm-scale channels are likely to be related to the initial planform conditions under which the experiments were conducted and the relatively small amount of time over which they were able to evolve (Fig. 11). These runs lasted approximately 30-35 minutes, and were constrained by the time it took for the upstream pond to melt through the ice block until water came out the bottom. In spite of this, the behavior observed, and values obtained are similar to those reported by other authors. In the case of Run 4 (Fig. 11, Table 2), sinuosity did not change significantly during the last ten minutes of the run. Measured values varied between 1.50 and 1.54, suggesting that approximate equilibrium conditions had been reached.

Leopold and Wolman (1960) observed that meandering took some time to develop after an irregular sheet of water became channelized in the Dinwoody Glacier. Shallow channels had no well-developed meandering pattern, but deeper channels with a width similar to that of the shallow channels showed well-developed sinuosidal patterns. As the channels incised vertically, sinuosity increased. This behavior matches what was observed in the experiments (Fig. 11) and the range of values measured, 1.03-1.54 with a median value of 1.36 (Table 1), is similar to the range 1.1-1.7 reported by Zeller (1967) for meltwater streams in Swiss glaciers. The values measured on the Peterman ice island also match these observations. The range of values observed is 1.25-1.95, with a median value of 1.50.

We believe that the differences observed in the meltwater channels are related to the ratio between vertical incision rates and lateral migration rates. We discuss this issue in section 4.2.

### 3.2.2 Wavelength-to-width ratio

The wavelength-to-width ratios measured for meltwater streams are shown in Figure 10b and Table 1. The meltwater streams have values in the range 5.0-13.1, with the smallest and largest values measured in the cm-scale runs. (Table 1). The range of measured values lies between what has been observed in alluvial rivers and other meltwater channels.





Karlstrom et al. (2013) plotted a compilation of sixty one wavelength vs. width pairs for meltwater meandering channels. Their values have the following statistics: Min. = 5.4; $q_1$ = 9.2; Median = 10.8; $q_3$ = 15.1; Max. = 37.6. The rivers in the NCHRP database have $q_1$ = 8.2; Median = 9.7; $q_3$ = 12.1. The median values for the mm-scale, cm-scale and ice island, vary between 6.8 and 7.9. Nevertheless, a larger sample would be needed to assess if the differences between the medians are

significant. All previous data refer to meltwater channels with $10^{-1}$ m < B < $10^{1}$ m; we have extended this to include channels with widths as small as $10^{-3}$ m.

### 3.2.3 Fatness and skewness

Figures 10c and 10d show boxplots of the coefficients of fatness and skewness (Fig. 4) which were determined using the routines developed by Vermulen et al. (2016). The method determined that 57% of the bends in the mm-scale channels are

round. In the case of the cm-scale and the ice island channels, half the bends were found to be round and half angular. The only meltwater channels showing preferential upstream skewness of their bends are the mm-scale ones, with 54% having a negative skewness coefficient. The bends in the cm-scale and ice island channels show preferential downstream skewness. Results for the NCHRP data show 53% of the bends are round ($f_f$ > 0) and 57% are upstream skewed ($f_s$ < 0).

In the context of alluvial rivers and in the absence of other information, upstream skewness was thought to be predominant

and therefore an indication of flow direction (Parker et al. 1982). The rivers in the NCHRP dataset show preference for upstream skewness but the ratio is not as dominant as to indicate flow direction based uniquely on planform images of channels. Moreover, the results presented herein suggest that in environments where meandering is not caused by sediment erosion and deposition, downstream-skewed bends are more common. An example from a different environment where downstream-skewed bends might be more dominant is shown in Figure 7d. It shows a short reach of a meandering channel in karst, created

by bedrock dissolution. Almost all bends therein are downstream-skewed.

The results in Table 1 and Figure 10 show that meltwater channels with lower sinuosity values have more downstream-skewed bends than otherwise, and that round bends are as common as angular bends. These results might be related to the absence of sediment deposition, which has been inferred to contribute to the development of upstream skewness and roundness of bends (Parker et al. 1982).

### 3.3 Flow properties, migration rates, and downstream trends in the cm-scale rivulets

Flow properties for the cm-scale runs are summarized in Table 2. Reported depth values were estimated from other known variables assuming a rectangular cross section at the straighter portions of the reach. The median widths and estimated depths were used to compute width-to-depth ratios for all channels. The values obtained vary between 3.2 and 13.5 with a median value of 6.3. Parker (1975) included a set of 7 channels from the Barnes Ice Cap with a median aspect ratio of 5.4, and Leopold

and Wolman (1960) reported on two channels with an aspect ratio of ~8.

Froude numbers were calculated as: $F_r = U(gH)^{-1/2}$ where g is acceleration of gravity (9.81 m/s$^2$). Reynolds numbers were calculated as follows: $R_e = UR_H\nu^{-1}$ where: $R_H = BH(B + 2H)^{-1}$ is an estimate of the hydraulic radius and $\nu$ is the kinematic





viscosity of water. Since the water temperature was changing from upstream to downstream, a value of $v = 1.4x10^{-6}\ m^2/s$, corresponding to a temperature of 7°C, was used. The reach-averaged velocities reported for Run 01 are estimations of the minimum value that could have prevailed in each case. The time between images for this run was 10 seconds, and therefore velocities could not be calculated as in the other runs, where the position of the dye front was tracked in subsequent images.

This affects the Froude and Reynolds numbers as well as the dimensionless migration rates. We decided to include these values in our compendium because they at least provide an order of magnitude.

The Froude numbers obtained at different times during runs 02, 03 and 04 show both supercritical and subcritical flow conditions (Table 2). Karlstrom et al. (2013) show that meandering is possible when Fr > ~0.4 and the channel aspect ratios are between 5 and 10. These conditions were also observed in our cm-scale experimental runs.

In all the cm-scale laboratory cases, the Froude numbers show a decreasing trend associated with a decrease in reach-averaged flow velocity, sinuosity growth, and decreasing channel slopes (Table 2). The decreasing trends observed in the downstream direction are related to the cooling of water as it flows downstream. As the temperature gradient between flow and channel boundary decreases, thermal erosion is less effective. Also, the rubber molds (e.g. Fig 9c) show larger vertical incision in the upstream portions than the downstream portions. We did not attempt to control this downstream reduction in temperature

gradient, which is likely difficult to do in the laboratory and might also exist in supraglacial channels.

Reynolds numbers ($R_e = UR_H v^{-1}$) computed for the different runs are in the laminar ($R_e < 575$) regime. All values are below 200. Our laboratory observations on meltwater meandering channels provide one more set of examples of fluvial morphodynamic features that are possible by the interaction of purely laminar flow with an erodible bed (Lajeunesse et al., 2010), an issue we discuss in Section 4.1.

Measured dimensionless lateral migration values vary between 1.6 x $10^{-4}$ and 4.9 x $10^{-4}$. In alluvial rivers, some lateral migration models compute bank erosion by using a dimensionless migration coefficient that varies between $10^{-6}$-$10^{-8}$ (e.g. Motta et al., 2012). Our experiments show migration rates that are at least two orders of magnitude larger than typical values used in numerical models of alluvial river meandering. This result was expected and is thought to be dependent on the temperature gradient between the flowing water and the ice boundary.

## 4 Discussion

Logistical challenges render field research in glacial environments difficult. In spite of the advances in numerical models and remote sensing capabilities (Pitcher and Smith, 2019), laboratory experiments are a viable and exciting complementary path to better understand the processes driving supraglacial meltwater meandering channel hydro- and morphodynamics. We have presented an initial step along this path.

We have successfully produced meandering rivulets over an ice-block in the laboratory. Observations on centimeter scale channels developing from a set initial channel were complemented with observations on incidental self-formed millimeter scale channels. Our experiments provide insight into the processes of formation and evolution of meltwater meandering



channels on ice. Both sets show morphologies that are typically observed in meandering channels in other media and at larger scales, but they also show differences worth discussing. Their significance, however, needs to be analyzed in light of the issue of scale before we can treat them as analogs of supraglacial meltwater channels instead of relatives.

## 4.1 Scaling (ir)relevance

Scaling is typically the first issue that comes to mind when trying to relate laboratory observations with field analogs. Depending on the problem at hand, dynamic similarity between model and prototype should be satisfied in order to relate the laboratory measurements to the field scale in a precise way. Nevertheless, in the case of earth surface processes, small scale laboratory experiments have an 'unreasonable effectiveness' (Paola et al. 2009) capable of providing valuable insights into the underlying physics, without necessarily matching all relevant dimensionless parameters.  The main aspects related to the scale
and setting of our experiments are flow regime, surface tension effects, and temperature.

### 4.1.1 Flow regime

Our experiments had Reynolds numbers well within the laminar regime, but this has been the case in many other studies looking at geomorphic processes. Examples include meandering channel processes (Smith, 1998), drainage basin evolution (Hasbargen and Paola, 2000), steady-state erosional landscapes (Bonnet and Crave, 2006), braided streams (Metivier and
Meunier, 2003), and sand bedforms in open-channel flow (Coleman and Eling, 2000). Many more examples are discussed by Paola et al. (2009) and Lajeunesse et al. (2010). In spite of being in the laminar regime, our meltwater channels compare qualitatively and quantitatively very well with field cases, where flow regimes are turbulent.

Lajeunesse et al. (2010) and Malverti et al. (2008) show that turbulence is not necessary to create the morphologies themselves and that the most important aspect is the convergence of the underlying physics. The former authors suggest that 'it is possible
for a pair of flows to be simply two manifestations of the same phenomenon, both of which are described by a shared physical framework'. Similar findings are presented by Paola et al. (2009) who suggest that 'unreasonable experimental effectiveness arises from natural scale independence'. The three studies provide convincing evidence that flow regime in and by itself is not an impediment to relate experimental observations to field analogs and that lessons learned at the small scale can be upscaled appropriately.

### 4.1.2 Surface tension

Malverti et al. (2008) also address the issue of surface tension. They conclude that surface tension is not an important aspect as long as the experimental scales are larger than the capillary length of the fluid ($l_c = \sqrt{\sigma \rho^{-1} g^{-1}}$ where $\sigma$ is surface tension, $\rho$ is density, and $g$ is gravitational acceleration), or as long as the channel Bond number (Eq. 2) is greater than unity. The Bond number is a measure of the ratio of gravity forces to surface tension force. Peakall and Warburton (1996) summarize some
empirically suggested guidelines for critical Weber numbers (Eq. 3) below which, surface-tension induced distortion may be



expected in experimental work involving small scale channels. The Weber number is a measure of the relative importance of inertial forces to surface tension force. Based on their work, the minimum critical value of the Weber number may be 10. The value, however, is empirically based.

$$B_0 = \rho g L^2 \sigma^{-1} \tag{2}$$

$$W_e = \rho U^2 L \sigma^{-1} \tag{3}$$

In Eq. 2 and Eq. 3, $L$ is a characteristic length scale of the flow. For water at 5°C in air, the capillary length is 2.8 mm. It is readily seen that surface tension effects might have been present in our mm-scale channels. Bond and Weber dimensionless numbers were computed for the average channel characteristics in the cm-scale experiments, but only the former was computed for the mm-scale experiments due to the lack of flow velocity measurements. The hydraulic radius ($R_H$) was the typical length scale used for the cm-scale channels, and the average channel width was used for the mm-scale channels.

Table 3 shows the parameters used to compute the dimensionless numbers and the results obtained. The results suggest that surface tension effects are important in both cases. The morphologies observed suggest otherwise. We do not deny their presence but from the bend shapes, we suggest they are not as dominant as these values indicate.

Rivulets over a plane can meander due to an instability associated with surface tension (e.g. Le Grand-Piteira et al., 2006; Schmuki and Laso, 1990; Davies and Tinker, 1984; Goricky, 1973). All these studies acknowledge some planform similarities between surface tension generated meanders and those of alluvial rivers. Nevertheless, the meanders in these rivulets are fixed in space and time as long as the slope of the plane is constant and the input flow is steady. Once established, surface tension meandering channels display completely stationary paths (Le Grand-Piteira et al., 2006), i.e. bends do not migrate or change shape in time. Goricky (1973) observed that sinuosity values did not grow above 1.5 in his experiments, and led him to conclude that meander bend cutoffs and oxbow lakes could not be simulated on the surface tension stream plate. The signatures of lateral and vertical incision observed in our experiments suggest that additional dynamic processes occurred and that surface tension effects were not dominant.

### 4.1.3 Temperature

Lajeunesse et al. (2010) show a comparison between point-bar morphodynamics in bend flow in a river (turbulent regime) and a laboratory channel (laminar). They show that even though the morphologic signatures are comparable in both scenarios, the smaller channel required a higher Froude number, and a much higher slope. Laminar river analogs have 1.5-2.5 higher slopes than their natural counterparts (Malverti et al., 2008). In the context of our experiments, such analyses suggest that temperature gradients in laminar flows might need to be greater than those in their turbulent analogs so as to recreate specific morphologies. All our observations were conducted in a purely melting environment with ambient temperatures well above the freezing point. We measured fluid and ice temperatures only during the cm-scale runs (Table 2). The temperature differences between water and ice were at least 17°C at the inlet and between 2.5-4°C at the outlet. It is likely that the cm-scale experiments had smaller Eckert, and larger Stefan numbers than those in supraglacial streams. It is also likely that the mm-scale meanders did have similar values to those observed in the field since the water in the mm-scale channels was meltwater from the ice block and



not inflow from an external source. Observed differences in planform characteristics between cm- and mm-scale rivulets (Table 1), could be related to the temperature gradients prevalent in both sets of experiments.

## 4.2 Planform and channel morphology differences

We have shown that the cm- and mm-scale meandering rivulets have features commonly observed in meandering channels in other media and at different scales. They also have some differences that raise an important question regarding our understanding of meandering channel processes and morphologies in ice. What variables exert control over sinuosity growth and bend skewness in meltwater meandering channels on ice?

Low sinuosity values measured in the cm-scale rivulets (compared to the mm-scale) are likely to be related to the ratio between vertical incision rate and lateral migration rate. In general, the more dominant lateral migration is with respect to vertical incision, the faster sinuosity is expected to grow. In addition, very high incision rates might suppress the formation of cutoffs by preventing two parts of a bend developing in 3D from intersecting each other. We infer that the ratio between vertical and lateral incision is a function of flow velocity and temperature difference between the flow and the ice.

Upstream skewness and roundness of bends is believed to be due to sediment deposition (e.g. Parker et al., 1982), and Marston (1983) observed differences in meander bend shapes when debris was present in the stream. In the absence of sediment, the analog to deposition in the meltwater channels would be recrystallization (freezing) of water back onto the ice surface. Based on the temperatures measured in the cm-scale experiments (Table 2) this was not occurring. Downstream skewness was preferentially observed in the cm-scale rivulets where no depositional analog was present. These conditions are also present in the image of meanderkärren (Fig. 7d) where bends grow by dissolution of karst and precipitation is either absent or not as dominant. In the case of the mm-scale it is possible that recrystallization was occurring thus creating conditions for bends to also show upstream skewness.

Daily temperature variations in glacial environments might provide sufficient conditions to create both erosional and depositional analogs, with melting being predominant during the day and recrystallization at night. Temperature gradients along the channels are a function of depth, solar radiation, and the presence or absence of sediment (Isenko et al., 2005). Conducting laboratory experiments with better temperature control might allow looking at these aspects in greater detail. An issue that would need to be considered in the laboratory is the connection between supraglacial and englacial channel analogs. Both kinds of channels are relevant in the context of climate change where longer summer melt periods are expected, and larger volumes of meltwater are expected to be transported by these channels (Irvine-Flynn et al., 2011).

## 5 Conclusions

Our experimental observations on cm- and mm-scale meltwater meandering channels in the laboratory lead to the following conclusions:

Earth **Surface**
Dynamics
Discussions

1. Small scale laboratory experiments are capable of reproducing morphologies observed in natural scale rivers even if dynamic similarity is not satisfied.

2. Specifically, small-scale meltwater meandering rivulets with laminar flow, have lateral migration signatures and planform morphologies also observed in meandering channels with turbulent flows.

3. Relatively slow sinuosity growth of meltwater meanders might be related to larger incision rates as compared to lateral migration rates. Channels that incise vertically at a faster rate have in some sense less time to migrate laterally, thus slowing down sinuosity growth.

4. Meltwater channels with larger temperature differences between the flow and the ice boundary showed preferentially downstream skewed bends. In the absence of ice recrystallization (freezing), no sediment deposition analogs exist
which would promote upstream bend skewness.

5. Ambient temperature control, which would lead to better constrained small-scale laboratory experiments, is necessary to assess long term equilibrium of meltwater channels. A key aspect for future research on meltwater rivulets is the relative importance of vertical vs. lateral incision rates and their effect on channel and planform morphologies.

**Code Availability**

No distributable code was developed in this study.

**Data availability**

All data used in preparation of this manuscript is available at https://doi.org/10.13012/B2IDB-4384362_V1

**Video supplement**

Videos showing two cm-scale runs are available at: https://doi.org/10.5446/40431 and
https://doi.org/10.5446/40432

**Author contributions**

Experiments were designed by both authors. R. Fernández conducted the experiments, data analysis and post-processing. Manuscript was prepared and edited by both authors.



**Competing interests**

The authors declare that they have no conflict of interest.

**Acknowledgements**

We would like to thank Colin P. Stark for insightful discussions during the experiments and data processing, and for valuable feedback on a previous draft of this manuscript. We would like to thank Jaclyn Daum, Eduardo Hanon, and John Berens for their contributions to this study. We would like to thank D. Kaszlikowski for his drone image of a supraglacial meander bend in Pakistan. We also thank Stephen Marshak for his image of meanderkärren, and for copious insightful discussion. Participation of both authors in this study was made possible thanks to funding provided by the US National Science Foundation [grant EAR1124482]. Participation of R. Fernández also possible thanks to the Leverhulme Trust, Leverhulme Early Career Researcher Fellowship [ECF-2020-679].

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



**Table 1 Statistics of planform parameters**

| Channel Type | Sinuosity $\Omega$ [ - ] / Wavelngth-to-Width Ratio $\lambda$/B [ - ] | | | | | | | | | | | Fatness | | Skewness | |
|---|---|---|---|---|---|---|---|---|---|---|---|---|---|---|---|
| | Min. | | $q_1$ | | Median | | $q_3$ | | Max. | | | $f_f > 0$ / $f_f < 0$ | | $f_s > 0$ / $f_s < 0$ | |
| mm-Lab | 1.50 / | 5.6 | 1.72 / | 7.5 | 2.01 / | 7.9 | 2.21 / | 9.1 | 2.79 / | 10.4 | | 57 / | 42 | 45 / | 54 |
| cm-Lab | 1.03 / | 5.0 | 1.24 / | 6.2 | 1.36 / | 6.9 | 1.46 / | 8.8 | 1.54 / | 13.1 | | 50 / | 50 | 53 / | 47 |
| Ice island | 1.25 / | 5.7 | 1.43 / | 5.9 | 1.50 / | 6.8 | 1.54 / | 8.1 | 1.95 / | 9.0 | | 50 / | 50 | 80 / | 20 |
| NCHRP Rivers | 1.14 / | 5.6 | 1.37 / | 8.2 | 1.67 / | 9.7 | 1.92 / | 12.1 | 2.92 / | 19.7 | | 53 / | 46 | 42 / | 57 |





Earth **Surface** Dynamics Discussions

**Table 2 Centimeter-scale experiment results**

| Run | Time | Median Width | Average Velocity | Estimated Depth | Width-to-depth ratio | Froude Number | Reynolds Number | Dimensionless Migration Rate | Average Slope | Temperature | | | Sinuosity |
|-----|------|-----|-----|-----|-----|-----|-----|-----|-----|-----|-----|-----|-----|
| | | | | | | | | | | Inflow | Outflow | Ice Block | |
| | t [min] | B [cm] | U [cm/s] | H [cm] | B/H [-] | $F_r$ [-] | $R_e$ [-] | x $10^4$ [-] | S [-] | T [°C] | T [°C] | T [°C] | $\Omega$ [-] |
| 01 | 0.0 | 1.2 | - | - | - | - | - | - | - | 5 | - | - | 1.03 |
| | 7.0 | 2.0 | 7 | 0.36 | 5.5 | 0.4 | 132 | 9.0 | - | 5 | - | - | 1.06 |
| | 20.0 | 2.6 | - | - | - | - | - | - | - | 5 | - | - | 1.09 |
| | 36.5 | 2.3 | 8 | 0.28 | 8.0 | 0.5 | 127 | 4.3 | - | 5 | - | - | 1.19 |
| 02 | 0.0 | 1.1 | - | - | - | - | - | - | 0.073 | 17 | - | - | 1.07 |
| | 4.6 | 1.5 | 25.0 | 0.11 | 13.5 | 2.4 | 171 | 1.6 | - | - | - | - | 1.14 |
| | 9.8 | 1.6 | 15.6 | 0.17 | 9.4 | 1.2 | 156 | 2.7 | - | 18 | - | - | 1.24 |
| | 12.4 | 1.8 | 13.9 | 0.17 | 10.6 | 1.1 | 142 | 2.8 | 0.053 | - | - | - | 1.34 |
| | 14.4 | 1.7 | 12.0 | 0.20 | 8.6 | 0.9 | 139 | 2.0 | - | - | - | - | 1.39 |
| | 18.8 | 2.0 | - | - | - | - | - | - | - | 19 | - | - | 1.35 |
| | 36.3 | 2.2 | - | - | - | - | - | - | 0.029 | - | - | - | 1.41 |
| 03 | 0.0 | 1.0 | 13.3 | 0.31 | 3.2 | 0.8 | 183 | - | 0.097 | 15.5 | 2 | -1.8 | 1.17 |
| | 4.7 | 1.3 | 17.2 | 0.18 | 7.4 | 1.3 | 175 | 4.9 | - | - | 2 | - | 1.24 |
| | 10.5 | 1.5 | 13.7 | 0.21 | 7.0 | 1.0 | 158 | 2.7 | 0.053 | 16 | 2 | -1.6 | 1.31 |
| | 15.2 | 1.6 | 14.8 | 0.18 | 8.6 | 1.1 | 155 | 2.4 | - | - | 2 | - | 1.41 |
| | 21.6 | 1.2 | 15.1 | 0.23 | 5.2 | 1.0 | 179 | 2.5 | - | 17 | 2 | -0.8 | 1.45 |
| | 28.8 | 1.5 | 10.1 | 0.27 | 5.5 | 0.6 | 145 | 2.2 | 0.058 | - | 2 | - | 1.47 |
| 04 | 4.0 | 1.3 | 12.2 | 0.27 | 4.6 | 0.8 | 166 | - | 0.062 | 14 | 2 | -1.8 | 1.31 |
| | 7.7 | 1.5 | 10.7 | 0.25 | 6.0 | 0.7 | 146 | 3.2 | - | - | 2 | - | 1.35 |
| | 11.1 | 1.6 | 12.5 | 0.21 | 7.9 | 0.9 | 147 | 2.9 | 0.056 | - | 2 | - | 1.37 |
| | 14.4 | 1.4 | 12.1 | 0.25 | 5.4 | 0.8 | 159 | 4.3 | - | 18 | 2 | -1.2 | 1.44 |
| | 18.8 | 1.4 | 10.1 | 0.29 | 5.0 | 0.6 | 148 | 2.4 | 0.036 | - | 2 | - | 1.50 |
| | 21.4 | 1.6 | 9.7 | 0.26 | 6.3 | 0.6 | 139 | 3.4 | - | 19 | 2 | -1.0 | 1.54 |
| | 24.1 | 1.3 | - | - | - | - | - | - | - | - | 2 | - | 1.53 |
| | 27.1 | 1.7 | 8.7 | 0.28 | 5.9 | 0.5 | 132 | - | - | - | 2 | - | 1.52 |
| | 28.8 | 1.6 | - | - | - | - | - | - | 0.025 | 21 | 2 | -0.6 | 1.54 |



Earth **Surface**
Dynamics
Discussions



2      **Table 3 Bond and Weber Numbers for the laboratory meandering channels.**

| Parameter | | mm-scale | cm-scale |
|---|---|---|---|
| B | [m] | 0.0007 | 0.015 |
| H | [m] | - | 0.0025 |
| $R_H$ | [m] | - | 0.0019 |
| U | [m/s] | - | 0.125 |
| $\rho$ | [kg/m$^3$] | 999.9 | |
| g | [m/s$^2$] | 9.81 | |
| $\sigma$ | [N/m] | 0.0749 | |
| $B_o$ | [ - ] | 0.06 | 0.46 |
| $W_e$ | [ - ] | - | 0.39 |



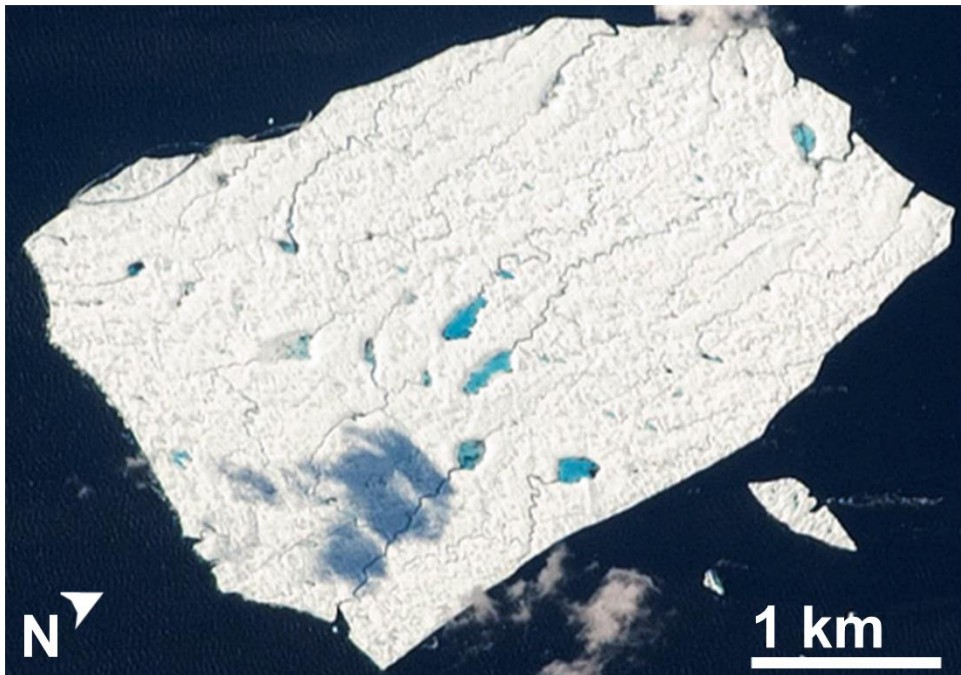

**Figure 1 Ice Island with melt ponds and meltwater channels on the Atlantic Ocean close to Newfoundland, Canada. Information: Image of Peterman Ice Island A, fragment 2 captured by an astronaut from the International Space Station on August 29th, 2011. Source: ISS Crew Earth Observations experiment and Image Science & Analysis Laboratory, Johnson Space Center.**





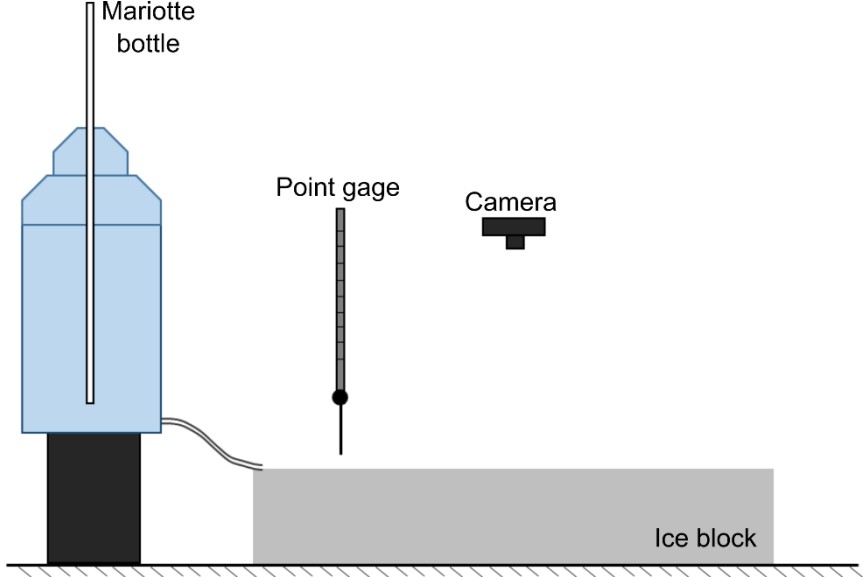

**Figure 2 Experimental setup for cm-scale laboratory meltwater meandering channels.**



Earth **Surface**
Dynamics
Discussions



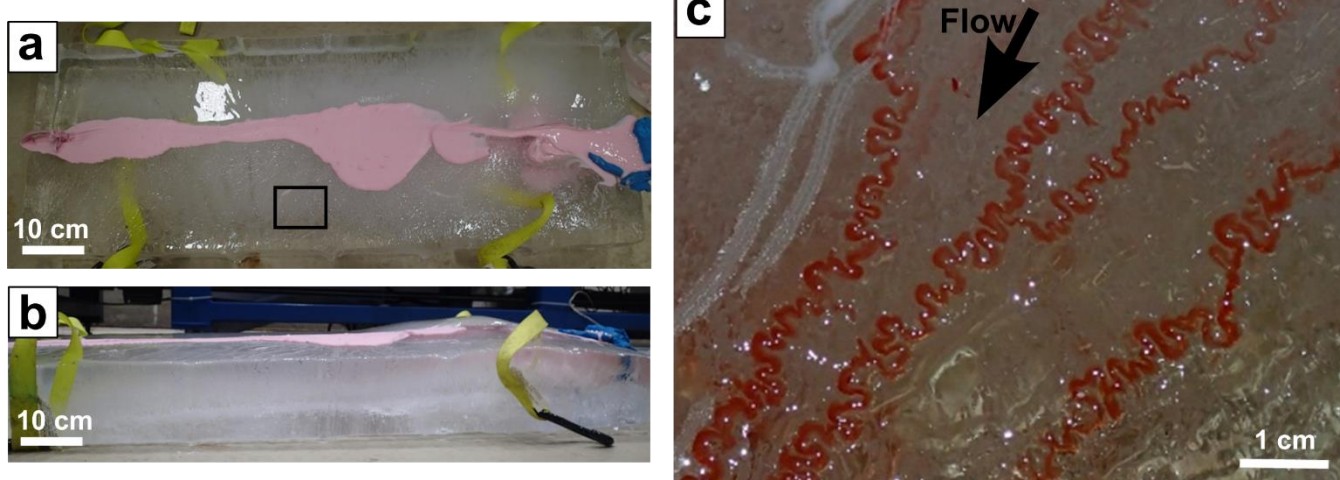

**Figure 3 (a) Top and (b) side view of ice block over which (c) first set of mm-scale self-formed meltwater meanders were observed. Area shown in (c) corresponds to region indicated in (a). Images in (a) and (b) were taken before adding dye.**





Earth **Surface**
**Dynamics**
Discussions

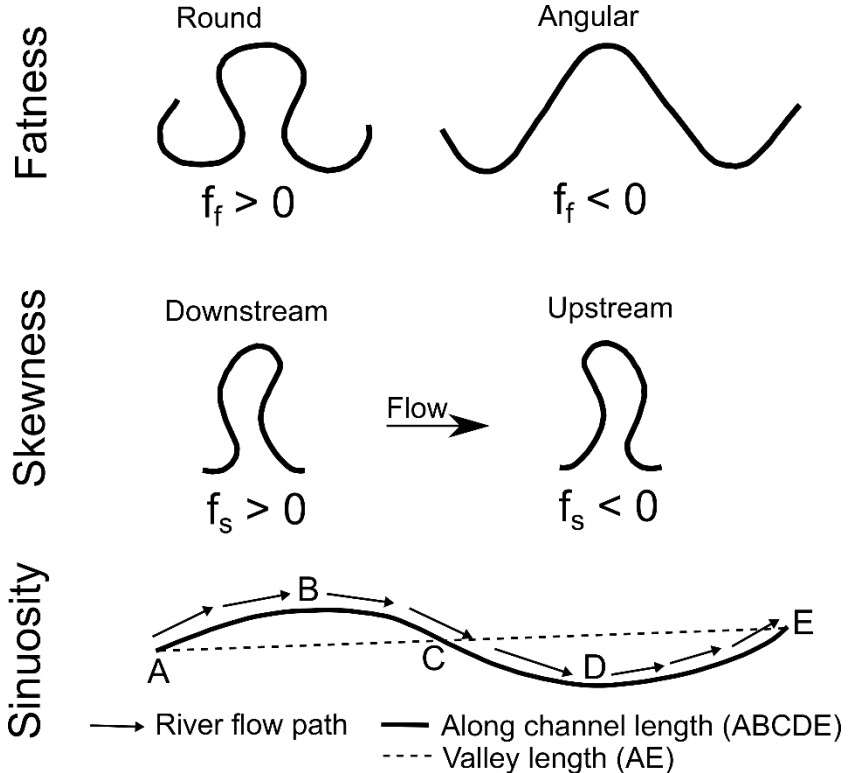

**Figure 4 Meander bend parameter definitions**



Earth **Surface**
Dynamics
Discussions


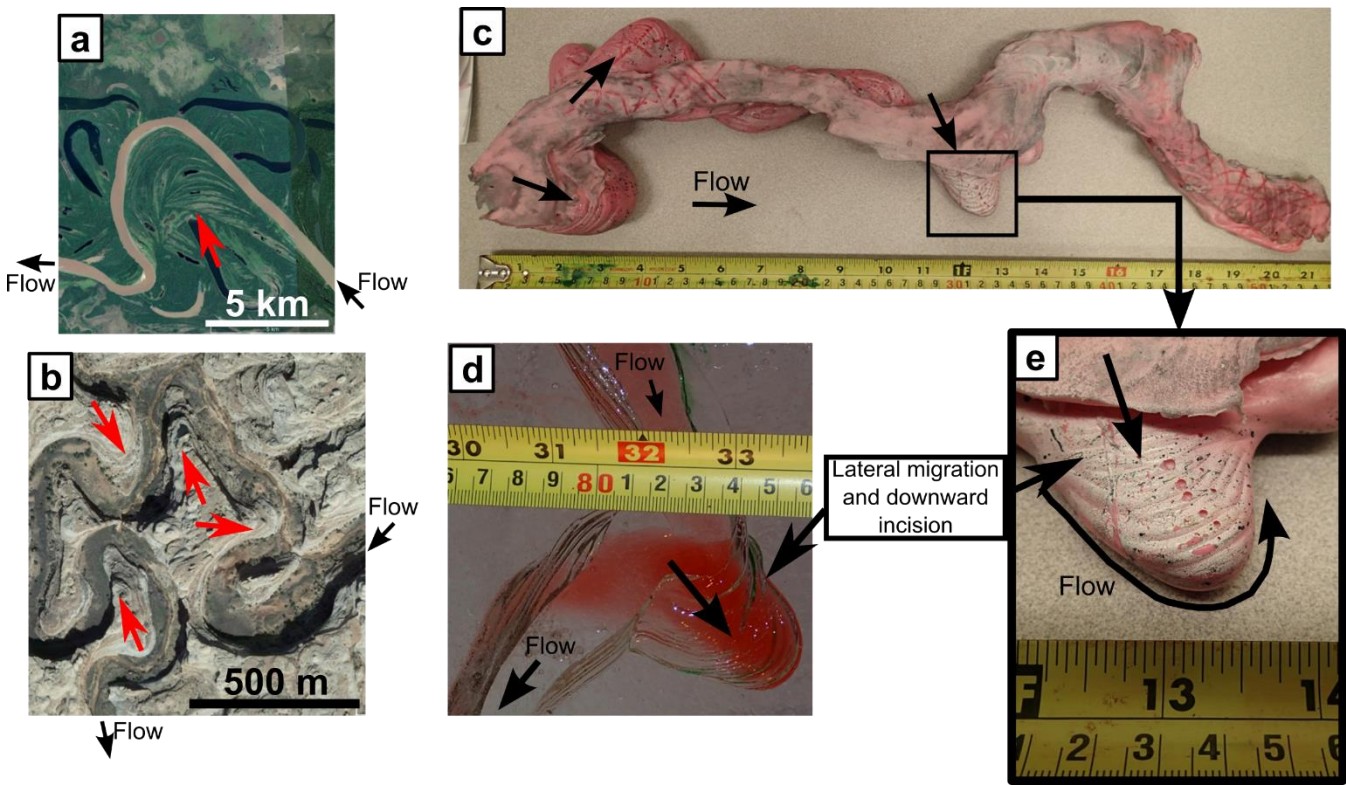

**Figure 5 Pulsed lateral migration features indicated by arrows in (a) an alluvial river (scroll bars); (b) a bedrock river; (c) rubber mold of a meltwater laboratory channel; (d) meltwater laboratory channel; and (e) close up of bend in rubber mold region indicated in (c). Image information: (a) Mamore river, Beni, Bolivia; Location: 255,863.55 m E 8,514,710.26 m S; Source: © Google Earth 2020. (b) Tributary of the San Juan River; Bedrock river in the Mogollon Rim, Utah, USA; Location: 574,447.88 m E 4,141,533.73 m N; Source: © Google Earth 2020. (c)-(e) Source, this study.**



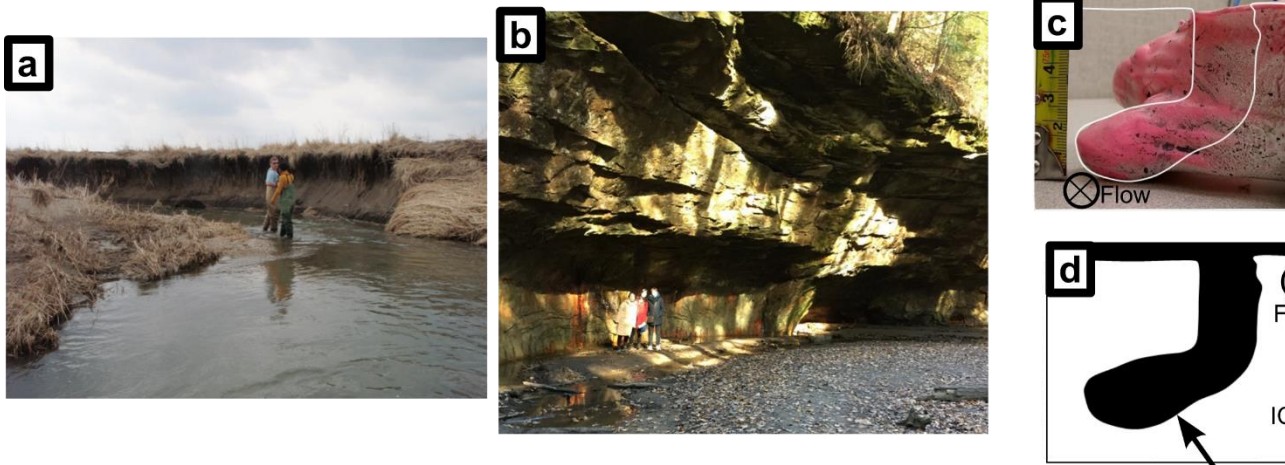

**Figure 6 Undercut bank in (a) an alluvial river; (b) a bedrock river; (c) a rubber mold of a laboratory meltwater channel and (d) a schematic of the laboratory meltwater channel in (c). Image information: (a) Embarras River, Indiana, USA. Picture by G. Parker; (b) Turkey Run, Indiana, USA. Picture by G. Parker; (c) Picture by R. Fernández.**



Earth **Surface** Dynamics
Discussions
EGU

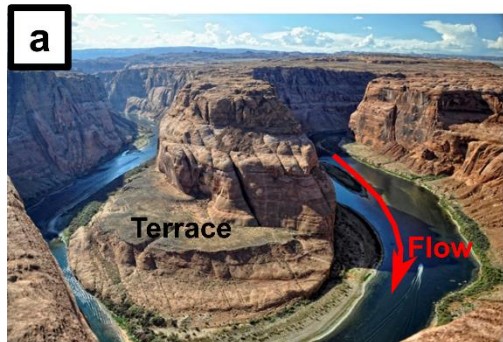

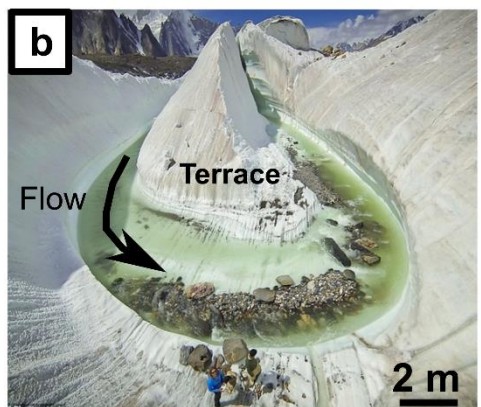

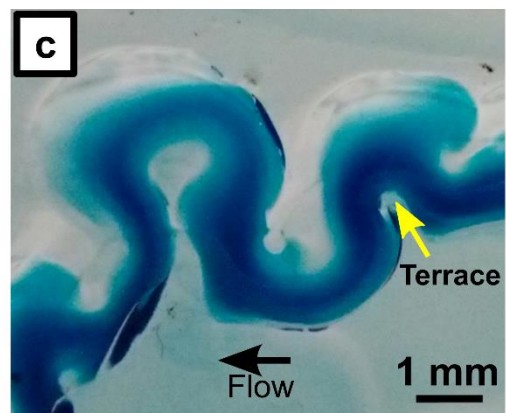

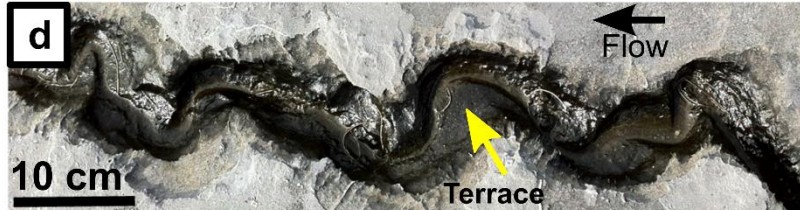

**Figure 7 Bend skewness and terrace in (a) a bedrock river, (b) a supraglacial meltwater meandering channel, (c) a self-formed mm-scale laboratory meltwater meandering channel and (d) soluble limestone (meanderkärren). Image information: (a) Horseshoe bend, Arizona. Source: Hermans P., 2012. (b) Concordia, Pakistan. Flow direction unknown. Source: D. Kaszlikowski. (c) Source: This study. (d) The Burren, Ireland. Source: S. Marshak.**



Earth **Surface**
**Dynamics**
Discussions

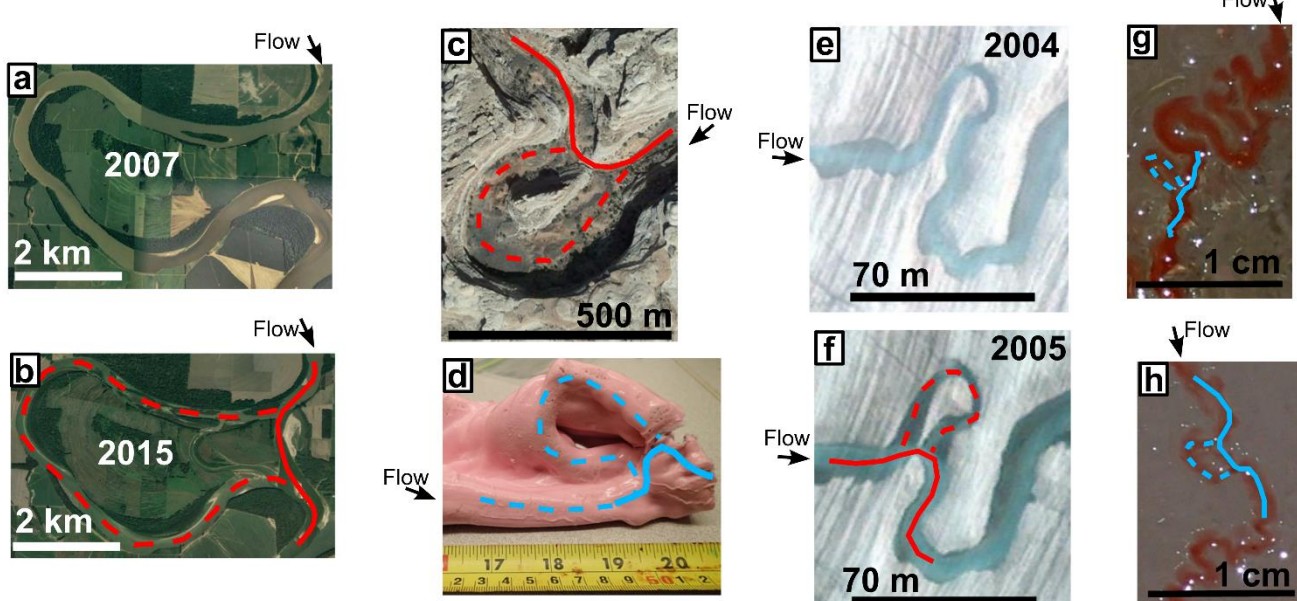

**Figure 8** Cutoffs in (a)-(b) an alluvial river, (c) a bedrock river; (d) a rubber mold of a laboratory melt water channel; (e)-(f) a supraglacial meltwater stream; and (g)-(h) self-formed meltwater laboratory channels. Image information: (a)-(b) Chute cutoff in a bend of the Wabash River just upstream of the confluence with the Ohio River; Location: 408,181.46 m E 4,186,022.16 m N; Source: © Google Earth 2020. (c) Bedrock tributary of the San Juan River; Mogollon Rim, Utah, USA; Location: 574,447.88 m E 4,141,533.73 m N; Source: © Google Earth 2020. (d) Neck cutoff on rubber mold from a laboratory meltwater channel. Source: this study. (e)-(f) Supraglacial meltwater stream in the Root Glacier, Alaska; Location: 399,186.46 m E 6,831,268.59 m N; Source: © Google Earth 2020. (g)-(h) Neck and chute cutoffs on self-formed laboratory meltwater meandering channels. Source: This study.





Earth **Surface**
Dynamics
Discussions

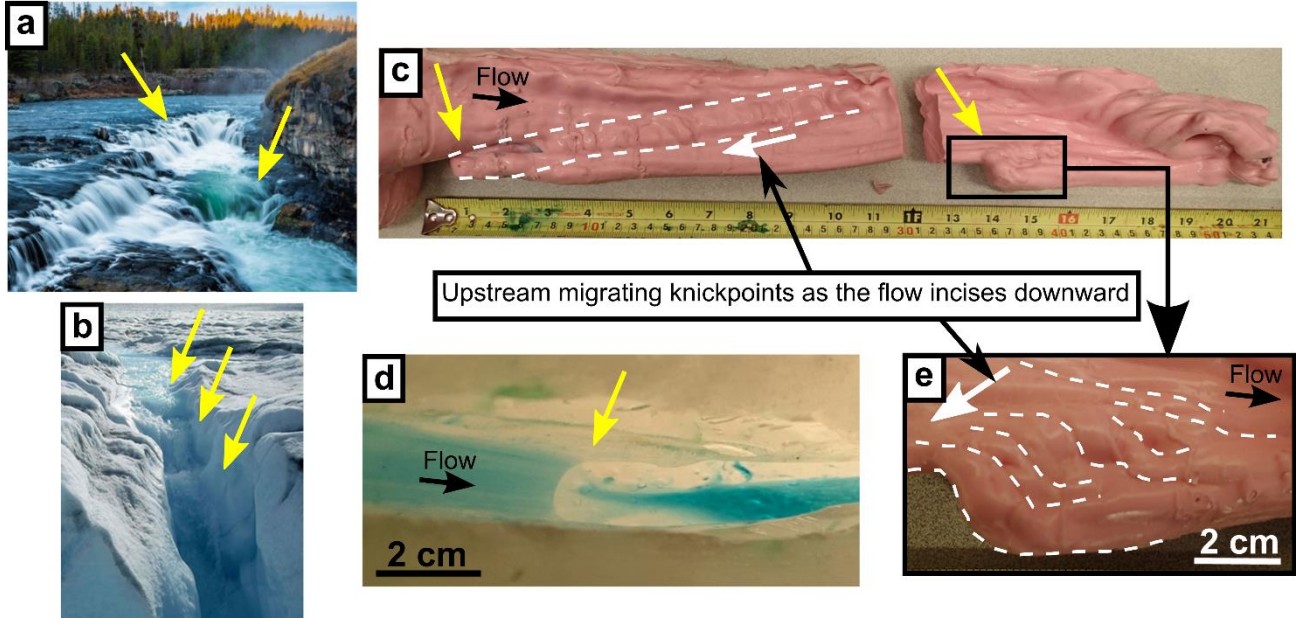

**Figure 9 Knickpoints indicated by yellow arrows on (a) mixed bedrock alluvial river, (b) supraglacial meltwater channel, (c) rubber mold of a laboratory meltwater channel, (d) laboratory meltwater channel and (e) close-up of rubber mold shown in (c). Image information: (a) Sheep Falls at Falls River, ID. (Source: J. Packer). (b) Supraglacial meltwater channel flowing towards a moulin on the Greenland ice sheet. (Source: J. Kastengren). (c)-(e) Source: this study.**







**Figure 10 (a) Sinuosity, (b) wavelength to width ratio, (c) fatness and (d) skewness of meltwater meandering channels and the**



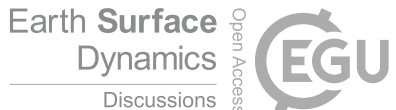

**rivers in the NCHRP (see text) database.**

**Figure 11 Planform evolution for four different cm-scale laboratory meltwater meandering channel experiments.**