# Peer review of "Laboratory observations on meltwater meandering rivulets on ice"

_Earth Surface Dynamics, 2020_

## Referee Comment (RC1) · Anonymous Referee #1 · 1 Dec 2020

**Review**
**Laboratory observations on meltwater meandering rivulets on ice**
**by**
**R. Fernández and G. Parker**

December 1, 2020

The paper presents observations in a laboratory experiment by the authors, in which a flow of warm water generates meandering channels on a block of ice. The authors find two types of channels. Major ones are a few centimeters wide, while smaller rivulets spontaneously form as the ice thaws. They compare the geometry of these channels to field observations, either on ice or in entirely different settings (alluvial rivers, bedrock rivers, karst). The experimental value of most of the quantities used to characterize the morphology of the channels, such as sinuosity, skewness, etc., are comparable to those measured in the field.

The paper is carefully written, and the text is easy to follow, despite the systematic use of the passive form, which often confuses the reader about whose experiment or observation the authors are referring to.

The major channels indeed exhibit many features that also exist in natural systems, and the authors present those in details. I must say, however, that I find the shape of the smaller channels more striking (Figure 3c). To me, these spontaneous, millimeter-scale channels resemble natural meanders more than the larger ones. I agree with the authors when they suggest that the former are probably closer to the thermal equilibrium than the latter, and I would think this closeness to equilibrium is essential to their developing a well-defined, single-wavelength instability. Based on this observation, I would be much interested to see what the major channels would look like were they fed with water barely above 0°C, and the experiment run for a long time.

The above suggestion, of course, is beyond the scope of the present manuscript which, I believe, can be published in E-Surf after the authors have addressed the minor points listed below.

**Writing and minor points**

- Page 4: "sometimes possess double-valued planforms" what does this mean? Why is sinuosity the only parameter that's mathematically defined? How about "fatness" and "skewness"? Please provide their mathematical definition.

- Figure 7: The legend says "(b) Concordia, Pakistan. Flow direction unknown.", but an arrow shows the flow direction on the picture nonetheless.

- Page 6: "It is likely that the neck cutoff produced a knickpoint": This could be argued for based on the amplitude of the knickpoint, the river's slope and the length of the loop before cutoff.

- Figure 10: The fatness seems not much different from zero, given the variability in measurements. This seems at odds with the pictures previously shown.

- Page 7: "[...] 2.01 whereas the latter have a median value of 1.36": How are these values "very different from each other" when the variability is about 25%?

- Figure 11: How come runs 3 and 4 are so similar their planform curves appear to match almost perfectly?

- Page 7: How do you measure the wavelength of the planform? There seem to be many wavelengths in the planforms of figure 11.

- Page 9, lines 12 and 24: "temperature gradient" → "temperature difference"

- Page 9, line 20: How do you make the migration speed dimensionless?

- Page 11, Please define mathematically the Bond and Weber numbers when they appear in the text. "The lack of flow velocity measurements": this is at odds with the previous estimation of a Reynolds number.

- The text about surface tension is confusing. Capillary rivulets on a solid surface are fixed in place by the hysteresis of the contact angle, a mechanism related to, be different from, surface tension. In the present experiment, surface tension most likely affects the shape of the water surface. This, however, doesn't mean that it is directly related to the (unknown) process by which the meandering instability occurs.

- Page 11: "Laminar river analogs have 1.5-2.5 higher slopes": some might, but there is no rule here. It is about 0.1 in some experiments [1] and $10^{-3}$ in others [2]. Similarly, there is no typical value for the slope of an alluvial river, since it scales like the inverse square root of its discharge, which varies over many orders of magnitude [3, 4].

- Page 11: "It is also likely that the mm-scale meanders did have similar values": Indeed, they are probably much closer to the thermal equilibrium, and their shape resemble natural meanders more strikingly. I believe this is a good reason to maintain the main channels as close to equilibrium as possible.

**References**

[1] P Delorme, O Devauchelle, L Barrier, and F Métivier. Growth and shape of a laboratory alluvial fan. *Physical Review E*, 98(1):012907, 2018.

[2] A Abramian, O Devauchelle, and E Lajeunesse. Laboratory rivers adjust their shape to sediment transport. *Physical Review E*, 102(5):053101, 2020.

[3] O. Devauchelle, A. P. Petroff, A. E. Lobkovsky, and D. H. Rothman. Longitudinal profile of channels cut by springs. *Journal of Fluid Mechanics*, 667:38–47, 2011. doi: 10.1017/S0022112010005264.

[4] F. Métivier, O. Devauchelle, H. Chauvet, E. Lajeunesse, P. Meunier, K. Blanckaert, P. Ashmore, Zh. Zhang, Y. Fan, Y. Liu, et al. Geometry of meandering and braided gravel-bed threads from the Bayanbulak Grassland, Tianshan, PR China. *Earth Surface Dynamics*, 4(1):273–283, 2016.

---

## Referee Comment (RC2) · Anonymous Referee #2 · 6 Dec 2020

This manuscript is concerned with a laboratory study of meltwater flow over an icy surface, which is intended to be an analogue for supraglacial flows of glaciological relevance. The authors are primarily concerned with the morphology of spontaneously incised drainage pathways arising in their experiments, which they compare quantitatively to similar morphologies observed in fluvial environments, finding many similarities. Then the authors go on to discussing how their mm-to-cm scale laboratory patterns differ from the much larger scale natural supraglacial channels, and argue that despite the difference in scale, their laboratory channels can be considered good proxies for the supraglacial ones, and therefore the insight obtained by comparison with the fluvial setting can be extended supraglacial meanders.

While the paper is nicely written, the data presented appear to be carefully analyzed,

and great care is taken to compare laboratory observations to datasets of fluvial morphologies, in my view the relevance of this work to supraglacial channels remains marginal. Besides some relatively minor comments on the Introduction (see minor points below), there are two key aspects of the experimental setup that motivate my point of view: the laminar regime, and the very warm (up to 21 deg C at the inflow, not kept constant over the course of each experiment, as per table 2) water temperature. Both these aspects affect the energy balance of the flow in ways that make hard for me to believe that any similarity may exist with the supraglacial setting: there meltwater is originated by melting of snow/ice, and is therefore very close to the melting point temperature (if not overcooled), so having enough energy available for melting comes down to turbulent heat dissipation in the flow. This is a very different configuration from the one studied in the laboratiry, where all the energy is supplied by the very warm water (which, incidentally, cools down by over 10 deg between inflow and outflow, demonstrating my point), with internal heating being negligible due the laminar regime. In light of this, unsteadiness in the inflow temperature (see table 2) may be significant in driving the meandering instability, and perhaps in producing some of the observed features. This should be at least acknowledged in the manuscript, and possibly elaborated on.

To counter my main criticism above, I encourage the authors to consider carefully these thermal aspects in their Discussion section, which I would recommend to rewrite with increased focus on the supraglacial (rather than fluvial) setting. Why did you choos water temperature to be this warm? Does it have any effect on the morphology? How does it compare to natural supraglacial setting? Can you tell us anything about heat fluxes at the ice water interface? These are all questions glaciologically inclined readers would want to see adreessed. At the moment, the applicability of this study to supraglacial streams comes across as an after thought. I believe that, upon a thorough revision, this paper may become as relevant to the glaciological community as is already to the fluvial morphology one, hence I would encourage the authors to pursue this angle at depth.

Minor points:

1) Introduction: I am not sure about what is special about the satellite image from the Petermann glacier brough as a motivation. There have been a number of similar observations of surface melt lately, in similarly or even more unexpected places (a good starting point for a literature review would be Kingslake et al. 2017, https://doi.org/10.1038/nature22049), which all show similar morphologies. As written, the paper suggests that the Petermann iceberg is somewhat special, which I think is deceiving.

More broadly, I find that the glaciological motivation (and literature) provided in the introduction is rather scant. At present, modelling glacier surface hydrology (beyond water routing models) remains very challenging for ice sheet modellers, mostly because the physics governing the topology of the network are not quite clear yet and hard to model. Experimental work can help constrain those physics, so why not to mention this aspect as well in your introduction?

2) Page 2, line 18: define supercritical and subcritical flow conditions

3) Page 3, line 14: what is a periodontal probe?

4) page 4, line 19: " direction spatial series was .." there is something wrong with the text here, please check

5) Page 12, lines 26-27: I am not sure why increased meltwater discharge due to climate warming should make these channels more relevant. Are the authors hinting at any particular physical process?

---

## Author Comment (AC1) · 13 Dec 2020

Response to Reviewer No. 1 Prepared by Roberto Fernández on behalf of both authors

Thanks for your positive review and your comments. We really appreciate your time. Please see below a preliminary response to your comments, which we hope to address fully in the coming days.

Page 4: "Double-valued planforms" refers to channels, which in a Cartesian plane have two 'y' coordinates for a single 'x' coordinate ($y = y(x)$ is not single valued).

"Meander bends typically show certain systematic deviations from simple Cartesian sinusoidal forms. Bends tend to be round and full, or 'fat', often to the point of possessing double-valued plan-forms, as Langbein & Leopold (1966) have noted." (Parker et al.

[Figure]

1982).

We will make sure to add the definitions for fatness and skewness. These terms come from the transformation of Langbein and Leopold (1966) intrinsic coordinate meander equation, to Cartesian coordinates. This transformation involves the generation of higher-order modes. The fatness and skewness coefficients are third order modes. See Parker et al. (1982) for details. In this manuscript, we use the forms derived from the wavelet analysis of Vermeulen et al. (2016).

Figure 7: Thanks for pointing this out. We forgot to remove the 'Flow direction unknown' bit in the legend after the photographer provided information regarding the flow direction.

Page 6: Are you able to expand on this comment? Are you questioning the statement or providing a recommendation?

Figure 10: This comes as no surprise. More than 80% of the fatness coefficients for the NCHRP alluvial rivers vary between -0.1 and 0.1 and have a median value that is very close to zero. Vermeulen et al. (2016) show results for four different rivers. Their distributions also have a median, which is close to zero, and the vast majority of the data lie between -0.1 and 0.1 (see Fig. 4 therein).

Page 7: There is indeed high variability but the largest measured sinuosity for the cm-scale channels is very close to the smallest measured sinuosity for the mm-scale channels. We will edit this paragraph to highlight this issue, avoid using the term 'very different', and acknowledge the high variability.

Figure 11: Well-spotted! Run 04 is almost a continuation of Run 03 but not exactly. After Run 03 finished, we put the ice block back in the freezer and attempted to add just enough water to cover the melt pond that formed upstream. In the process, some water made it into the channel and modified the shape and slope enough that we decided not to consider it the same run. We will include this clarification in the manuscript.

Page 7: Each of the centerlines shown in Figure 11 was analyzed following the approach described in section 2.3 (p.7 Lines 19-24).

Page 9 (lines 12 and 24): Yes! We will change gradient to difference.

Page 9 (line 20): Migration rates were made dimensionless with the average flow velocity. We will add this information to the methods section.

Page 11: The Bond and Weber numbers are defined when they appear in the text. They first appear in the last paragraph of page 10 and the definitions are included right after this paragraph at the beginning of page 11. We only report velocity measurements (and Reynolds numbers) for the cm-scale channels. The 'lack of flow velocity measurements' refers to the mm-scale channels.

Surface tension: Thanks for your comment. We will make sure to edit this section to make it clearer.

Page 11: Thanks for these references. We will take them into account for this section and add clarification on the different slopes observed in other experiments.

Page 11: Yes! This is essential and I am working to achieve that. Stay tuned.

---

## Author Comment (AC2) · 13 Dec 2020

Response to Reviewer No. 2 Prepared by Roberto Fernández on behalf of both authors December 13, 2020

Thanks for your thought-provoking review, respectful criticism, and your comments on the manuscript. We really appreciate your time. Please see below a preliminary response to your comments, which we hope to address fully in the coming days.

Indeed, we come to the topic with a fluvial morphology interest and, overall, a motivation to understand the mechanisms that create such similar meandering planforms in spite of the broad range of scales and types of media. Our intent was first to see if we could create such channels in the laboratory and after achieving this we decided to do

multiple runs and compare the results with meandering channels in other media.

As you rightfully recognize, we come to the problem with a fluvial motivation. However, the incidental mm-scale channels are likely better analogs (not scaled models) of supraglacial channels than the cm-scale ones for reasons we acknowledge and that you point out (e.g. large temperature differences, laminar flow regime).

In answer to your questions in paragraph 2, page C2: We did not choose the water temperature but simply used tap water. In one of the experiments, we added ice to control the temperature but this proved impractical without a cold room (a facility we did not have). Water temperature does have an effect on the morphology. The cm-scale channels show preferential downstream skewness and smaller sinuosity values. Natural supraglacial settings must indeed be colder and temperature differences smaller than those observed in the cm-scale channels. The mm-scale channels however must have had temperatures closer to freezing. We did not look into heat fluxes at the ice-water interface. We believe this is certainly something that needs further experimentation in a better controlled environment. We see this manuscript as a very small but significant step towards linking the fluvial aspects that brought us to the topic with the glacial aspects that would broaden the applicability of this work to the glaciology community. We might not be able to offer in-depth analysis of the aspects most relevant to glaciologists now but your criticism is very valid and is something we have thought off for ongoing work on the topic. Be assured that current work does involve glaciologists and we hope to strengthen this link to provide better insights for both the fluvial and glaciology communities.

  In response to the minor points: 1) The picture of the Peterman ice island fragment was what triggered this effort. We only intend to acknowledge the fact that it was our main motivation and it led to our first trial runs. We will include a bit more motivation for the manuscript addressing the challenges facing the glaciology community and emphasizing the need for experimental work.

2) We did not consider this definition necessary because we wrote the manuscript with the fluvial perspective. We will include it to make sure readers from other communities have the definition at hand. For the time being:

Channelized flows can be subcritical, critical or supercritical depending on the value of the ratio between inertial forces and gravitational forces. This ratio is expressed with the Froude number $Fr = u / (g*H)$, where u is the flow velocity, g is the acceleration of gravity, and H is the flow depth. If the ratio is smaller than one ($Fr < 1$) the flow is subcritical; if the ratio is larger than one ($Fr > 1$) the flow is supercritical; and if the ratio is equal to one ($Fr = 1$), the flow is critical.

3) A periodontal probe is a tool used by dentists to measure the depth of the pockets between patients' gums and their teeth (My father is a dentist). Its tip is marked (every mm) and is narrow enough (< 1mm) that we could use it for the experiments without affecting the flow conditions too much.

4) The text is not wrong but we will modify it to make it clearer. 'Direction spatial series' is probably meandering river community jargon.

5) Those lines refer to the potential links between supraglacial and englacial channels mentioned in the previous sentence. We believe that increased meltwater production will lead to increased/altered links between such features and is something we hope to address in future experimental work (in a properly temperature controlled environment).

---

## Author Response (AR1)

*Response to Reviewers and Manuscript with Track Changes*
*Prepared by Roberto Fernández on behalf of both authors*
*December 26, 2020*

5 We would like to thank Jens Turowski for handling this manuscript and to both anonymous reviewers for the time spent providing thoughtful reviews to our manuscript. We have considered their comments, questions, and recommendations carefully and have attempted to provide a point-by-point response to all issues raised, first in the open discussion forum and now in this document.

10 Following up from the preliminary responses posted in the online discussion forum, we provide further clarification and include references to locations where edits have been made. We also include a version with track changes below (line and page numbers included below refer to the updated manuscript and not the version included below).

Given the characteristics of the reviews, we feel very positive about the current version of the manuscript and hope
15 the Associate Editor and Editor agree to it being promptly published in ESurf.

Reviewer 1:

**Page 4:** "Double-valued planforms" refers to channels, which in a Cartesian plane have two 'y' coordinates for a single 'x' coordinate (y = y(x) is not single valued).

20 "Meander bends typically show certain systematic deviations from simple Cartesian sinusoidal forms. Bends tend to be round and full, or 'fat', often to the point of possessing double-valued plan-forms, as Langbein & Leopold (1966) have noted." (Parker et al. 1982).

We have added the definitions for fatness and skewness and updated Figure 4 **(see Page 4, Lines 19-28 and Page 25).** These terms come from the transformation of Langbein and Leopold (1966) intrinsic
25 coordinate meander equation, to Cartesian coordinates. This transformation involves the generation of higher-order modes. The fatness and skewness coefficients are third order modes. See Parker et al. (1982) for details. In this manuscript, we use the forms derived from the wavelet analysis of Vermeulen et al. (2016).

**Figure 7:** Thanks for pointing this out. We forgot to remove the 'Flow direction unknown' bit in the legend after
30 the photographer provided information regarding the flow direction. We have updated the legend and this is not mentioned anymore. **See Page 28.**

**Page 6:** Are you able to expand on this comment? Are you questioning the statement or providing a recommendation? We did not fully understand this statement by R1.

**Figure 10:** This comes as no surprise. More than 80% of the fatness coefficients for the NCHRP alluvial rivers vary between -0.1 and 0.1 and have a median value that is very close to zero. Vermeulen et al. (2016) show results for four different rivers. Their distributions also have a median, which is close to zero, and the vast majority of the data lie between -0.1 and 0.1 (see Fig. 4 therein). **Figure 4 (Page 25)** shows the effect of fatness coefficients on a Kinoshita generated (**Eq. 1 – Page 4**) meander bend. Values used therein are ±0.033 and ±0.01.

**Page 7:** There is indeed high variability but the largest measured sinuosity for the cm-scale channels is very close to the smallest measured sinuosity for the mm-scale channels. We will edit this paragraph to highlight this issue, avoid using the term 'very different', and acknowledge the high variability. We have edited this paragraph. **See Lines 21-24, Page 7.**

**Figure 11:** Well-spotted! Run 04 is almost a continuation of Run 03 but not exactly. After Run 03 finished, we put the ice block back in the freezer and attempted to add just enough water to cover the melt pond that formed upstream. In the process, some water made it into the channel and modified the shape and slope enough that we decided not to consider it the same run. We will include this clarification in the manuscript. **See Page 3, Lines 11-15.**

**Page 7:** Each of the centerlines shown in Figure 11 was analyzed following the approach described in section 2.3 (**Page 4**).

**Page 9 (lines 12 and 24):** Yes! We will change gradient to difference. This change has been made in the locations where it was pertinent. **See Page 9, Line 26, Line 29; Page 10, Line 5; Page 12, Line 13, Line 18, Line 29; Page 13, Line 23.**

**Page 9 (line 20):** Migration rates were made dimensionless with the average flow velocity. We added this information to the methods section. **Page 5, Line 13.**

**Page 11**: The Bond and Weber numbers are defined when they appear in the text. They first appear in page 11 and the definitions are included right after this paragraph (Eq. 3 and 4). We only report velocity measurements (and Reynolds numbers) for the cm-scale channels. The 'lack of flow velocity measurements' refers to the mm-scale channels. **Page 11, Lines 11-19.**

**Surface tension**: Thanks for your comment. We will make sure to edit this section to make it clearer. We have edited this section slightly to acknowledge the likely presence of surface tension but indicate that it is not directly related to the instability triggering the meandering process. **See Page 11 Line 25-26 and Page 12, Line 3.**

**Page 11:** Thanks for these references. We will take them into account for this section and add clarification on the different slopes observed in other experiments. We have added references in the text to acknowledge the different range of slope values observed in the laminar regime by others. **See Page 12, Lines 8-9.**

**Page 11**: Yes! This is essential and I am working to achieve that. Stay tuned.

Reviewer 2:

In addition to the responses provided during the open discussion, we have added edits to the manuscript in response to the respectful criticism and feedback offered by R2.

We have made an acknowledgement in the manuscript indicating that we focus on fluvial aspects and believe that this study is a first step in a longer and necessary effort to link fluvial geomorphology and glaciology. **See Page 2 Line 32 - Page 3 Line 2.**

We have added the definition of the Froude number. **See Page 2, Lines 18-20.**

We have added a description of a periodontal probe. **See Page 3, Line 21-22.**

We have made edits to section 2.3 to clarify the meaning of the 'direction spatial series'. The inclusion of Eq. 1 and its explanation (**Page 4, Line 21-28**) as well as references to specific sections of the supplemental material (**Page 5, Lines 1-2**) should allow all readers to understand the meaning of the direction spatial series of a meandering channel.

We have edited the closing sentence of the discussion to remove the confusing statement and strengthen our message about the importance of conducting experiments to better understand supraglacial meltwater channel dynamics in conjunction with other tools such as remote sensing and numerical modeling. **See Page 13, Lines 8-12.**

[revised manuscript text omitted]

---

## Referee Report (RR1)

**2nd review**
**Laboratory observations on meltwater meandering rivulets on ice**
**by**
**R. Fernández and G. Parker**

January 18, 2021

I find the second version of the manuscript significantly clearer than the first. I am therefore supportive of publication in E-Surf at this point.

Regarding the comment about the knickpoint, (formerly page 6): my point was a suggestion for improvement. The text says "It is likely that the neck cutoff produced a knickpoint"; I believe this can be supported—or invalidated—by measurements. If this hypothesis is correct, then the amplitude of the knickpoint, say $\Delta h$, should be related to the river's slope, $S$, and to the length of the bend before cutoff, $L_b$, through:

$$\Delta h = SL_b. \tag{1}$$

Is this relation satisfied in the experiment, at least in order of magnitude?

Finally, about the slope of laminar rivers (formerly page 11). The new manuscript is still misleading about this. The sentence "Laminar river analogs have been observed to have 1.5-2.5 higher slopes than their natural counterparts (Malverti et al., 2008), suggesting a wide range of possibilities in the laminar regime" is correct, strictly speaking, but suggests that the factor of 1.5-2.5 is typical. It is not, because there can be no such thing as a typical slope for a river (laminar or natural). The slope of a river, in first approximation, is related to its discharge through a power law, and of course there is no typical discharge for a river. In fact, this factor could be pretty much anything, given enough variety for the fluid viscosity, and a sufficiently powerful pump for the experiment.

---

## Author Response (AR2)

**Author's Response**

**Prepared by Roberto Fernández on behalf of both authors**

We would like to thank the AE Jens Turowski and the two anonymous reviewers for their feedback on the manuscript.

The 'track changes' version of the manuscript shows a lot of changes but in reality the text did not change that much. Based on feedback from the AE we moved text from Results to Discussion and this created lots of changes in the manuscript.

Specific changes made, worth mentioning are:

1.  We modified the four aspects pointed by the AE in terms of language.

In response to Reviewer 1:

2.  We added the following text (now p6 L1-4)

*The knickpoint height ($\Delta h$) in Figures 9c and 9e is approximately 1cm and the meander bend length before the cutoff ($L_b$) in Figure 8d is approximately 14cm long. These two variables are related via the channel slope as $\Delta h = SL_b$. The slope for this run is not available but the knickpoint height divided by the meander bend length is equal to $S = 0.077$, within the range of average slopes obtained in the cm-scale experiments (Table 2).*

3.  We modified the order of the sentence regarding the laminar channel slopes to make sure that the reader first sees that 'a wide range of slope values are possible', hopefully no longer being a misleading statement.

*In the laminar regime, a wide range of slope values are possible. For example, laminar river analogs have been observed to have 1.5-2.5 higher slopes than their natural counterparts (Malverti et al., 2008) and others have also reported values such as 0.1 (Delorme et al., 2018) and $5x10^{-3}$ (Abramian et al., 2020).*

---

## Author Response (AR3)

**Author's Response**

**Prepared by Roberto Fernández on behalf of both authors**

We would like to thank the editor Niels Hovius, AE Jens Turowski and the two anonymous reviewers for their feedback on the manuscript.

In attention to the AEs last suggestions:

1.15 grammar: something missing here, maybe '…which is in spite…'?
12.13 parentheses should include 'below', i.e., (Eq. 4 below)

We have not modified 1.15. We believe that the sentence is correct. In the manuscript we discuss the relevance (or irrelevance of scale) of our experiments which in spite of being in the laminar regime… shed light on certain processes.

We have modified 12.13 for clarity. The word 'below' refers to Weber numbers below which surface tension effects might be important. We have modified the text as follows:

Peakall and Warburton (1996) summarize some empirically suggested guidelines for critical Weber numbers (Eq. 4). The authors suggest values below which, surface-tension induced distortion may be expected in experimental work involving small scale channels.

Thanks again for your very valuable feedback and time!